# Extracellular Vesicles Isolated from Plasma of Multiple Myeloma Patients Treated with Daratumumab Express CD38, PD-L1, and the Complement Inhibitory Proteins CD55 and CD59

**DOI:** 10.3390/cells11213365

**Published:** 2022-10-25

**Authors:** Kieran Brennan, Katrine F. Iversen, Alfonso Blanco-Fernández, Thomas Lund, Torben Plesner, Margaret M. Mc Gee

**Affiliations:** 1School of Biomolecular & Biomedical Science, University College Dublin (UCD), Dublin 4, Ireland; 2Conway Institute of Biomolecular and Biomedical Research, University College Dublin (UCD), Dublin 4, Ireland; 3Institute of Regional Health Science, University of Southern Denmark, 7100 Vejle, Denmark; 4Department of Internal Medicine, Section of Hematology, Lillebaelt Hospital, University Hospital of Southern Denmark, 7100 Vejle, Denmark; 5Flow Cytometry Core Technology, UCD Conway Institute of Biomolecular and Biomedical Research, University College Dublin (UCD), Belfield, Dublin 4, Ireland; 6Department of Hematology, Odense University Hospital, 5000 Odense, Denmark

**Keywords:** daratumumab, extracellular vesicles, plasma, bone marrow, multiple myeloma, complement, EV biomarkers, CD147

## Abstract

Daratumumab (DARA) has improved the outcome of treatment of multiple myeloma (MM). DARA acts via complement-dependent and -independent mechanisms. Resistance to DARA may result from upregulation of the complement inhibitory proteins CD55 and CD59, downregulation of the DARA target CD38 on myeloma cells or altered expression of the checkpoint inhibitor ligand programmed death ligand-1 (PD-L1) or other mechanisms. In this study, EVs were isolated from peripheral blood (PB) and bone marrow (BM) from multiple myeloma (MM) patients treated with DARA and PB of healthy controls. EV size and number and the expression of CD38, CD55, CD59 and PD-L1 as well as the EV markers CD9, CD63, CD81, CD147 were determined by flow cytometry. Results reveal that all patient EV samples express CD38, PD-L1, CD55 and CD59. The level of CD55 and CD59 are elevated on MM PB EVs compared with healthy controls, and the level of PD-L1 on MM PB EVs is higher in patients responding to treatment with DARA. CD147, a marker of various aspects of malignant behaviour of cancer cells and a potential target for therapy, was significantly elevated on MM EVs compared with healthy controls. Furthermore, mass spectrometry data suggests that MM PB EVs bind DARA. This study reveals a MM PB and BM EV protein signature that may have diagnostic and prognostic value.

## 1. Introduction

Multiple myeloma (MM) is an incurable malignancy of the B-cell lineage, characterized by neoplastic, monoclonal expansion of plasma cells in the bone marrow (BM) [1]. Remarkable progress has been made in the treatment of MM with the introduction of immunomodulatory drugs, proteasome inhibitors, and most recently monoclonal antibodies [2,3,4]. However, despite this progress, MM remains an incurable disease.

Daratumumab (DARA) is a CD38 antibody approved for the treatment of MM as monotherapy or in combination with a number of standard of care anti-myeloma drugs. DARA induces direct killing of tumour cells via complement-dependent cytotoxicity (CDC), antibody-dependent cell mediated cytotoxicity (ADCC) and antibody-dependent cellular phagocytosis (ADCP) [5]. In addition, DARA affects the tumour microenvironment (TME) and the immune system in several different ways. DARA inhibits the adhesion of myeloma cells to stroma, inhibits the formation of nanotubes that transfer mitochondria from stromal cells to myeloma cells, depletes CD38+ regulatory cells of the T-, B- and M-phenotype and stimulates cytotoxic T-cells directly leading to the expansion and activation of cytotoxic T cells [6,7,8,9]. Furthermore, DARA prevents the formation of immunosuppressive adenosine by CD38 [10]. Over time during treatment with DARA the expression of CD38 is reduced and the expression of complement regulatory molecules CD55 and CD59 is increased, which may impair the tumour-killing activity of DARA by CDC, ADCC and ADCP [11,12]. The reduced expression of CD38 by myeloma cells during treatment with DARA is proposed to occur due to redistribution and release of CD38 on extracellular vesicles (EVs) rather than preferential killing of myeloma cells expressing high levels of CD38, because it occurs almost instantaneously after initiation of therapy [11,13]. Despite the many anti-tumour effects of DARA, the majority of patients eventually relapse. Real world data have revealed that the median overall survival for patients progressing on DARA is approximately one year, but this may change with the introduction of novel therapies targeting B-cell maturation antigen (BCMA) and other targets on myeloma cells [14,15,16]. 

The expression of CD38 on MM cells decreases rapidly after the first infusion of DARA in both responders and non-responders [11]. A high level of expression of CD38 before the initiation of DARA-treatment is associated with a better chance of response, but it does not result in a longer PFS [[11] and personal communication]. Preclinical studies revealed that all-trans retinoic acid (ATRA) increases the expression of CD38 on MM cells, which enhanced the DARA-mediated ADCC and CDC [17]. In the clinical setting, the addition of ATRA to DARA-treatment showed limited activity [18]. Whether or not the expression level of CD38 is important for resistance to DARA is still a matter of debate [19]. 

EVs are a heterogenous population of particles, ranging in size from 50 to 5000 nm and enclosed by a lipid bilayer, that are released from cells and play an important role in cell–cell communication in physiological and pathophysiological situations [20]. EVs can be subdivided into apoptotic bodies and microvesicles which are formed and released from the plasma membrane, with sizes ranging from 100–1000 nm (microvesicles) and 1–4 µm (apoptotic bodies), while exosomes have a diameter of 30–150 nm and are formed by inward budding of late endosomes/multivesicular bodies and are secreted when these multivesicular compartments fuse with the plasma membrane [21,22]. EVs are secreted in significantly higher amounts by cancer cells compared to normal cells [23]. These tumour-derived EVs mediate intercellular communication between tumour cells and normal cells within the tumour microenvironment via the horizontal transfer of functional protein, DNA and RNA [24,25,26]. Extracellular vesicles may also act as a decoy that captures and neutralizes therapeutic antibodies [27]. 

There is no difference in the pre-DARA-treatment levels of the complement inhibitory proteins CD55 and CD59 expression on MM cells when comparing responders to non-responders, but there is an upregulation of CD55 and CD59 at the time of progression during DARA monotherapy [11]. This may contribute to resistance to complement mediated cytotoxicity and the loss of response to DARA. CD55 on MM cells restricts the deposition of C4b and C3b, while CD59 inhibits complement membrane attack complex formation, and may thereby protect MM cells from DARA-mediated CDC [28,29]. Another potential mechanism of resistance involves programmed death ligand 1 (PD-L1), which is a ligand for the checkpoint receptor programmed death 1 (PD-1) expressed by several immune cells including T-cells [30]. The binding of PD-L1 to PD-1 on T cells induces T-cell apoptosis and anergy of tumour-specific T cells and thus resistance to T-cell mediated anti-myeloma activity [31,32]. It has also been shown that PD-L1 expressing MM cells are more resistant to apoptosis induced by melphalan and the proteasome inhibitor bortezomib [33,34]. PD-L1 may be expressed by tumour cells, antigen-presenting cells and multiple other cells of the body but importantly also by EVs [35]. Verkleij et al. showed that the expression of PD-L1 is significantly higher on MM cells from patients progressing on DARA compared to newly diagnosed multiple myeloma patients (NDMM) [36]. Malavasi et al. showed that EVs released from BF01 myeloma cells in vitro express PD-L1 and the complement inhibitory receptors CD55 and CD59, but the relevance of this observation to the situation in vivo is not clear [13]. 

The aim of this study was to isolate, quantify and characterize EVs from peripheral blood (PB) and bone marrow (BM) from MM patients treated with DARA and compare DARA-responders with non-responders. We found that the DARA target CD38, the complement inhibitory proteins CD55 and CD59, and the checkpoint ligand PD-L1 are present on MM-derived EVs. Furthermore, we reveal that the amount of PD-L1 on the EVs correlates with MM patient response to DARA. We also identified DARA in the EV samples by mass spectrometry suggesting DARA may be bound to the EVs through CD38.

## 2. Materials and Methods

### 2.1. Study Population and Sample Collection

Fifty-seven patients diagnosed with MM according to the IMWG guidelines and treated with a DARA-containing regimen at the Departments of Haematology at either Vejle Hospital or Odense University Hospital, Denmark participated in the study [37]. Nineteen of these patients were responding to treatment with DARA and 38 were progressing. Additionally, four patients with untreated, newly diagnosed multiple myeloma (NDMM) and twelve healthy subjects were included as controls. The heathy donors were matched by age and sex with the MM population. Participation was voluntarily, and written informed consent was obtained from all subjects. Samples were obtained between December 2019 and May 2021. Data on patient characteristics and number of prior treatment lines were retrospectively obtained from the electronic medical flies and registered in a designated Research Electronic Data Capture (REDCap) database [38,39]. The study was approved by The Regional Committees on Health Research Ethics for Southern Denmark (S-20170212). Platelet free plasma (PFP) was obtained by centrifuging peripheral blood (PB) and bone marrow (BM) aspirate samples two times at 2500× *g* at 4 °C, for 15 min. Samples were stored at −80 °C until EV isolation. 

### 2.2. Antibodies and Reagents

Flow cytometry antibodies; anti-CD9-PE (1–25, clone M-L13, 555372 BD Bioscience), anti-CD63-PE (1–100, clone H5C6, 556019 BD Bioscience), anti-CD81-APC (1–25, clone JS-81, 551112 BD Bioscience), anti-CD147-APC (1–400, clone MEM-M6/1, A15706 Thermo Fisher Scientific (Waltham, MA, USA)), anti-CD38-FITC (1–200, clone CYT-38F2, 1911229 CYTOGNOS), anti-CD55-BV750 (1–200, clone A10, 750101 BD Bioscience), anti-CD59-APC (1–200, clone OV9A2, 17-0596-42 Thermo Fisher Scientific (Waltham, MA, USA)), anti-PD-L1-PE-CY7 (1–100, clone MIH1, 558017 BD Bioscience), IgG1 isotype control-FITC (1–125, clone MOPC-21, 554679 BD Bioscience), IgG2 -BV750 (1–200, clone G155-178, 553456 BD Bioscience).

Western blot primary antibodies: rabbit anti-TSG101 (1:500, clone EPR7130(B), ab125011 Abcam), mouse anti-CD63 (H5C6-BD bioscience, 556019, 1/500), mouse anti-APOE (1/1000, clone f.9, sc-390925 Santa Cruz Biotechnology (Dallas, TX, USA)), mouse anti-APOA1 (1/1000, clone A5.9, sc-13549 Santa Cruz Biotechnology (Dallas, TX, USA)), mouse anti-Albumin (1/1000, clone AL-01, sc-51515 Santa Cruz Biotechnology (Dallas, TX, USA)), mouse anti-APOB (1/500, clone F2C9, MA5-14671 Thermo Fisher Scientific (Waltham, MA, USA)). Western blot secondary antibodies: anti-Rabbit IgG-DyLight 800 (1:5000 dilution, Thermo Fisher Scientific (Waltham, MA, USA), SA5-35571), anti-Mouse IgG-DyLight 680 (1:5000 dilution, Thermo Fisher Scientific (Waltham, MA, USA), 35519)

### 2.3. EV Isolation and Iodixanol Density Gradient Separation

All ultracentrifugations were performed in Beckman Coulter rotors and ultracentrifuge tubes at 120,000× *g* AVG in Beckman Coulter Optima L-100 XP or Beckman Coulter Optima MAX-XP ultracentrifuges, with centrifugation durations based on a “50 nm cut-off size” adjustment to the centrifugation duration for each rotor as described in Livshits et al. 2015, with additional 5 min added to allow the rotor to come up to speed [40]. 

#### 2.3.1. Peripheral Blood (PB) Platelet Free Plasma (PB PFP)

PB PFP samples were defrosted and diluted with particle-free PBS before centrifugation at 2500× *g* for 15 min in an SX4250 rotor to pellet cell fragments and other debris. The supernatant was transferred to a 38 mL ultracentrifuge tube (Prod. No. 344058). The tubes were centrifuged at 120,000× *g* (RCF AVG, 31,300 rpm) for 2 h 40 min at 20 °C, using a SW32ti rotor. The supernatant was removed and the EV pellet was resuspended in 1 mL residual PBS and transferred to a 1 mL ultracentrifuge tube (Prod. No. 343778). The tubes were centrifuged at 120,000× *g* (RCF AVG, 51,000 rpm) for 50 min at 20 °C, using a MLA130 rotor. The supernatant was removed and the EV pellet was resuspended in 200 µL residual PBS.

#### 2.3.2. Bone Marrow (BM) Aspirate Platelet Free Plasma (BM PFP)

BM PFP samples were defrosted and diluted with particle-free PBS before being centrifuged at 2500× *g* for 15 min in an SX4250 centrifuge to pellet cell fragments and other debris. The supernatant was transferred to a 13 mL ultracentrifuge tube (Prod. No. 344059). The tubes were centrifuged at 120,000× *g* (RCF AVG, 31,300 rpm) for 2 h 45 min at 20 °C, using a SW41ti rotor. The supernatant was removed and the EV pellet was resuspended in 200 µL residual PBS.

#### 2.3.3. Iodixanol Density Gradient Centrifugation

Density gradient centrifugation was performed using a modified protocol from Brennan et al. [41]. A 54% iodixanol-PBS working solution (estimated density ~1.295 g/mL) was prepared by diluting a stock solution of OptiPrep™ (60% (*w*/*v*) aqueous iodixanol from Axis-Shield PoC, Norway) with 10× particle-free PBS (Gibco, Waltham, MA, USA). Iodixanol solutions (1.2 g/mL and 1.08 g/mL) were prepared by diluting the 54% iodixanol-PBS working solution in 1× particle-free PBS (Gibco, Waltham, MA, USA). To form the gradient, firstly a homogenous base layer of the gradient (estimated density ~1.224 g/mL) was produced by adding 672 µL of the 54% iodixanol-PBS working solution to a 13 mL ultracentrifuge tube (Prod. No. 344059), together with 200 µL either BM or PB PFP EVs isolated by ultracentrifugation. Next, 2 mL 1.2 g/mL iodixanol and 3 mL 1.08 g/mL iodixanol were layered successively on top of the vesicle suspension with the remainder of the tube filled with PBS. Centrifugation was performed at 197,120 g (RCF AVG) for 15 h at 4 °C in a SW41ti rotor (40,000 rpm). Fractions (~200 µL) were collected from the top of the tube. 50 µL of each fraction was pipetted into a 96 well plate and absorbance was measured at 340 nm against an iodixanol standard curve to determine the fraction density. The fractions with densities between 1.08–1.19 g/mL were combined and diluted to a density <1.03 g/mL with particle-free PBS and the diluted fractions were centrifuged at 120,000× *g* (RCF AVG, 31,300 rpm using a SW32ti rotor) for 3 h 15 min at 20 °C. The supernatant was removed and the EV pellets were resuspended in 200 µL residual PBS and stored at −80 °C prior to analysis.

### 2.4. Flow Cytometric Analysis

#### 2.4.1. EV Detection and Counting

Flow cytometry analysis was performed on the Beckman Coulter CytoFLEX LX Flow Cytometer using a modified protocol from Wu et al. with a VSSC gain = 300; VSSC-H threshold = 5500 [42]. Events were gated on the VSSC-width log x VSSC-H log cytogram to remove EV aggregates (singlet gate). A rectangular gate was set on the VSSC-H log x RSSC-H log cytogram containing the 80 nm and 500 nm bead populations and defined as ‘PS beads 80 nm–500 nm gate’ followed by a “stable time gate” set on the time histogram in order to identify the microparticle region (Appendix A).

#### 2.4.2. EV-Bead Conjugated Flow Cytometry

1.25 × 10^7^ EVs/test was mixed with 0.2 µL/test aldehyde/sulfate latex beads (4 μm; Thermo Fisher Scientific, Waltham, MA, USA) in 200 µL PBS rotating overnight at 4 °C, with beads without EVs being used as a negative control (Appendix A). To block non-specific protein binding to beads 200 µL 2% BSA (2% BSA, 2 mM EDTA, 0.1% sodium azide in PBS) was then added to the samples to a final volume of 400 µL for 1 h at RT, followed by 45 µL of 1 M glycine for 30 min at RT. The samples were then centrifuged at 5500× *g* for 5 min, the supernatant was removed and the beads were resuspended in 100 µL PBS and 2 µL FC block was added for 10 min at RT. The samples were then centrifuged at 5500× *g* for 5 min and washed with 500 µL PBS three times. The beads were resuspended in 1% BSA 100 µL/test and aliquoted into fresh tubes. The beads were stained with antibodies for 30 min on ice and then centrifuged at 5500× *g* for 5 min and washed with 500 µL PBS three times. The samples were then resuspended in 200 µL PBS and flow cytometry analysis was performed on the Beckman Coulter CytoFLEX LX Flow Cytometer. Gating of EV-decorated 4 μm diameter beads was performed based on FCS/SSC parameters, so that unbound EVs or possible antibody aggregates are excluded from the analysis.

### 2.5. SDS-Polyacrylamide Gel Electrophoresis (SDS-PAGE) and Western Blot Analysis

Equal volumes of the gradient fractions were mixed with 4× Laemmli buffer (750 mM Tris-HCl pH 6.8, 5% SDS, 40% glycerol and 80 mM DTT) and heated to 95 °C for 5 min. Protein was resolved on 12% (6% for APOB detection) polyacrylamide resolving gels was performed using a modified protocol from Brennan et al. [41].

### 2.6. Transmission Electron Microscopy and Nanoparticle Tracking Analysis

Transmission electron microscopy and nanoparticle tracking analysis was performed as previously described in Wu et al. [42].

### 2.7. Mass spectrometry

For MS analysis, 2.5 × 10^8^ EV isolates (~0.8 µg protein) were resuspended in 6 M urea, 50 mM Tris-HCl, reduced and alkylated using dithiothreitol (8 mM final concentration) and iodoacetamide (20 mM final concentration). Then, samples were diluted to 1 M urea using 50 mM Tris-HCl and digestion was continued overnight by the addition of sequencing grade modified trypsin (Promega, Madison, WI, USA, 1.5 µg trypsin/EV sample). Following trypsin digestion, the samples were cleaned using C18 HyperSep SpinTips (Thermo Fisher Scientific, Waltham, MA, USA) and each sample analysed in duplicate on a Bruker timsTof Pro mass spectrometer connected to an Evosep One liquid chromatography system. Tryptic peptides were resuspended in 0.1% formic acid and each sample was loaded onto an Evosep tip. The Evosep tips were placed in position on the Evosep One, in a 96-tip box. The autosampler is configured to pick up each tip, elute and separate the peptides using a set chromatography method (30 samples a day). Each sample was eluted from its Evotip onto a 15 cm, 150 µm i.d. analytical column packed with 1.9 µm C18 AQ reverse phase media (V1106 Analytical Column NT—30 samples/day, Evosep). Peptides were delivered to the analytical column in buffer a (lcms grade water/0.1% formic acid) and were separated with an increasing buffer B gradient (lcms grade acetonitrile/0.1% formic acid) over 44 min a flow rate of 0.5 µL/min. The mass spectrometer was operated in positive ion mode, with a capillary voltage of 1500 V, dry gas flow of 3 L/min and a dry temperature of 180 °C. All data was acquired with the instrument operating in trapped ion mobility spectrometry (TIMS) mode. Trapped ions were selected for ms/ms using parallel accumulation serial fragmentation (PASEF). A scan range of (100–1700 *m/z*) was performed at a rate of 5 PASEF MS/MS frames to 1 MS scan with a cycle time of 1.03 s [43].

#### Data Analysis

The raw data was searched against the *Homo sapiens* subset of the UniProt/Swiss-Prot Reviewed/DARA FASTA sequence using the search engine MaxQuant (release 2.0.1.0) using specific parameters for trapped ion mobility spectra data dependent acquisition (TIMS DDA). Each peptide used for protein identification met specific MaxQuant parameters. Specifically, only peptide scores that corresponded to a false discovery rate (FDR) of 0.01 were accepted from the MaxQuant database search. The normalised protein intensity of each identified protein was used for label free quantitation (LFQ).

The data was searched in parallel with the proprietary software Peaks X+ (25/10/2019, Bioinformatic Solutions, Waterloo, ON, Canada) using specific parameters for trapped ion mobility spectra data dependent acquisition (TIMS DDA). Each peptide used for protein identification met specific parameters, i.e., only peptide scores that corresponded to a false discovery rate (FDR) of 1% were accepted from the Peaks database search.

### 2.8. Statistical Analysis

The medium fluorescent intensity (MFI) of each marker on the EVs is expressed as a median. The data sets were tested using the Mann–Whitney U test. *p* values of less than 0.05 were considered statistically significant. All statistical analysis were performed using Stata version 16.0 for PC (Stata Corp LP, College Station, TX, USA).

## 3. Results

### 3.1. Patient Characteristics

Peripheral blood (PB) from 38 patients was obtained at the time of progression on a DARA-containing regimen [44]. From 19 of these “non-responders” a corresponding bone marrow (BM) aspirate was also obtained. Additionally, PB samples were obtained from 19 patients responding to a DARA-containing regimen with a partial response (PR) or better (“responders”), and matching PB and BM samples were also collected from four patients with newly diagnosed multiple myeloma (NDMM). The non-responders had received a median of 4 (range 0–17) prior lines of therapy, the responders had received a median of 2 (range 1–5) prior lines of therapy. For further details, see Table 1.

### 3.2. Expression of EV Markers on MM PB and BM EVs

EVs with a density of 1.08–1.2 g/mL were isolated from peripheral blood (PB) and bone marrow (BM) aspirate samples by density gradient ultracentrifugation. This density range was chosen to allow for the isolation of the widest range of EVs while also reducing soluble protein and lipoprotein particle contaminants. Appendix A shows that using density gradient ultracentrifugation the EV markers TSG101 and CD63 are enriched in lane 5 the expected density range for EVs (1.08–1.2 g/mL), while albumin is most abundant at the bottom of the gradient in lanes 7 and 8. Furthermore, the LDL marker APOB is present only in lane 3 and is absent from the EV fraction (lane 5), while the markers of the other lipoprotein particles APOA1 and APOE are spread across the gradient fractions, but not enriched in the EV fraction. NTA and TEM analysis of a representative sample identified particles in the expected size range (Appendix A).

The isolated EVs from each PB and BM sample were counted using the CytoFLEX and events within the 80–500 nm polystyrene beads region were used for analysis as indicated in Appendix A. This EV count was used to normalise the patient EV samples and ensure the beads were coated with and equal number of EVs from each patient. Results reveal that all EV samples analysed were positive for the four EV markers CD9, CD63, CD81 and CD147 (Figure 1). While the amount of CD9, CD63 and CD81 varied among patients, there was no significant difference in the amount of CD9, CD63 and CD81 between PB EVs from healthy controls and MM patients (Figure 1). However, in contrast, the amount of CD147 on EVs was elevated in MM PB EVs relative to healthy PB EVs (Figure 1), which is not surprising given that CD147 is upregulated in MM cells [46,47].

### 3.3. Expression of CD38, CD55, CD59 and PD-L1 on MM PB EVs

Next, we examined the amount of the DARA target CD38, complement inhibitory proteins CD55 and CD59 as well as PD-L1 on EVs from MM patients and healthy individuals. CD38 was detected on all EV samples and there was no difference in the amount of CD38 on EVs from healthy PB vs. MM PB (Figure 2). There was no significant difference in the level of CD38 (MFI) on PB EVs between patients responding to DARA treatment (median MFI = 1325.7) and non-responders (median MFI 1076.4), *p* = 0.24 (Figure 3). EVs from the PB of newly diagnosed multiple myeloma (NDMM) patients had a lower level of CD38 (median MFI = 610.45), but not significantly different from patients who had received DARA, *p* = 0.07, however it should be noted that the sample size of NDMM is only four patients.

The complement inhibitory proteins CD55 and CD59 were present at significantly higher levels on EVs from MM PB compared to healthy PB (Figure 2). Similar to CD38, there were no significant difference in the median MFI of CD55 and CD59 when comparing EVs from the PB of responders (median MFI of CD55 = 600.2; CD59 = 253.9) to non-responders (median MFI of CD55 = 755.05; CD59 = 223.4), *p* = 0.21 for CD55 and 0.18 for CD59 (Figure 3). The MFI of both CD55 and CD59 was higher in patients that received DARA (i.e., responders and non-responders), compared to NDMM (Figure 4).

All samples were positive for PD-L1 (Figure 2), however, while there was no overall difference in the amount of PD-L1 on EVs from healthy PB vs. MM PB (Figure 2), PD-L1 was significantly higher on EVs from responders (median MFI = 770) compared to non-responders (median MFI = 193.7), *p* = 0.002 (Figure 3). There was no significant difference between DARA-treated patients and NDMM (Figure 4).

### 3.4. Comparison of MM PB and BM EVs

We next compared the amount of the EV markers and CD38, PD-L1, CD55 and CD59 between matched MM PB and BM EVs. However, the medium fluorescent intensity level of each EV marker was lower for all the BM EVs relative to the PB EVs (Appendix A), suggesting that the difference in background salts, proteins and particles between BM and PB EV samples is affecting the EV quantification. To overcome this and compare matched BM and PB EV samples the MFI levels of each EV marker were normalized to each individual patient’s average CD9 and CD63 intensity. CD9 and CD63 were chosen as their median MFI was similar between healthy and myeloma EV samples (Figure 1) and the inclusion of two EV markers would reduce the variability compared to the use of either marker alone. After normalisation there was no difference in CD63 or CD81 between EVs from PB of MM patients compared to BM (Figure 5). CD9 was significantly lower in BM EVs relative to PB EVs, while CD147 was significantly higher in the BM EVs relative to PB EVs. There was no difference in the levels of CD38 or PD-L1 on EVs from MM PB and MM BM (Figure 5). CD59 was significantly higher in the BM EVs relative to PB EVs, while CD55 was significantly lower in the BM EVs relative to PB EVs (Figure 5).

### 3.5. DARA Is Present in MM PB EVs

Due to non-specific binding or cross reaction of an anti-DARA antibody, flow cytometry could not be used to determine whether DARA was bound to the surface of MM EVs. To overcome this problem LC-MS/MS was performed on 10 DARA treated MM EV (5 responding to DARA and 5 progressing on DARA) and 10 untreated healthy control EV samples. The peptide sequences were compared with the Homo sapiens subset of the Uniprot Swissprot Reviewed/DARA FASTA sequence with the proprietary software Peaks X+ (Bioinformatic Solutions) using specific parameters for trapped ion mobility spectra data dependent acquisition (TIMS DDA). The peptides identified in the 10 untreated healthy controls matched between 80 and 92% of the DARA light chain and between 35 and 51% of the DARA heavy chain sequence, with the amino acids identified underlined in Appendix A. This high sequence similarity with the IgG background is unsurprising as DARA is a human IgG. We sought to detect DARA-specific sequences in the MM samples, therefore the sequences detected that are in common to the IgG background were first excluded from the analysis. A number of DARA peptide sequences were only detected in MM patient EVs and are listed for each patient in Table 2, with the unique amino acids underlined. Results reveal that 9 out of 10 MM patient’s EVs contain a peptide sequence that was not detected in the 10 control samples, with several MM patients having multiple DARA-specific peptides.

Qualitative analysis was performed in parallel against the Homo sapiens subset of the Uniprot Swissprot Reviewed/DARA FASTA sequence using the search engine Maxquant (release 2.0.1.0) using specific parameters for trapped ion mobility spectra data dependent acquisition (TIMS DDA). Appendix A shows that 222 proteins were identified in healthy control EVs and 252 in MM EV samples (234 in responders and 240 in non-responders). The majority of proteins were common to each group, with 199 proteins in common between MM and healthy EV samples. When compared to the vesiclopedia top 100 protein list of EV associated proteins, it was found that 40 of the top 100 proteins were detected, with 33 EV associated proteins detected in all 10 healthy control and MM patient samples tested (Appendix A). Figure 6 highlights that CD59 and CD147 are more abundant in MM EVs relative to EVs from healthy individuals, and is consistent with the flow cytometry findings (Figure 1 and Figure 2). Furthermore, the T cell marker CD8A was found to be significantly higher in MM EVs relative to healthy controls and in responders relative to patients progressing on DARA (Figure 6).

## 4. Discussion

Various studies have reported the possible contribution of EVs released by tumour cells to the generation of therapeutic resistance, suppression of the immune system and promotion of cancer progression [27,48,49,50]. These tumour EVs can enter the circulation and have potential as a minimally invasive way to diagnose and monitor disease. However, it is not known whether circulating plasma EVs from MM patients can reflect the phenotype of myeloma cells residing in the BM that are usually characterized following a BM aspirate and/or biopsy. In this report, we define the level of the EV markers; CD9, CD63, CD81, CD147 as well as CD38, CD55, CD59 and PD-L1 on PB EVs and BM EVs from MM patients treated with DARA and healthy control PB EVs. Malavasi et al. showed that the binding of DARA to CD38 on the cell surface results in the redistribution of CD38 into EVs from human myeloma cell lines, suggesting that CD38 levels on EVs may change during DARA treatment with the highest levels likely to be found following initial infusion of DARA [13]. We found that the MFI of the DARA target CD38 on EVs from patients who had received DARA was higher compared to EVs from NDMM, although this was not statistically significant which may be due to a low number of NDMM. Alternatively, it is possible that CD38 levels on EVs change during treatment and a better understanding will require longitudinal analyses.

CD38+ EVs may play a role in the development of drug resistant via binding DARA in circulation and thereby preventing it from interacting with the CD38 on MM cell surface. Ciravolo et al. found that HER2+ EVs bound the anti-HER2 therapeutic Trastuzumab, rendering breast cancer cells resistant to further Trastuzumab treatment, and a similar observation has been made for CD20+ EVs in lymphoma [27,49]. In this study, using mass spectrometry, we investigated the possibility that MM EVs can bind DARA and our results reveal that DARA is present on PB EVs isolated from patients who have received DARA treatment. This data raises the possibility that CD38+ EVs contribute to DARA resistance by acting as a decoy receptor for DARA preventing it from binding to CD38 on the cell surface.

In this study, we show that PD-L1 is enriched on the surface of MM EVs, which is consistent with previous reports in other cancer types including melanoma, breast cancer and prostate cancer EVs [31,51,52]. Furthermore, EV PD-L1 has been reported to recapitulate the effect of cell surface PD-L1 by directly binding to PD-1 on T cells and has a vital function in immunosuppression and tumour progression [53]. PD-L1 may be constitutively overexpressed as a result of PTEN deletions, PI3K/AKT mutations, EGFR mutations, MYC overexpression, or CDK5 disruption [54,55,56,57,58]. However, PD-L1 expression is also induced in tumour cells in response to interferon-γ released by activated T cells [59]. Therefore, in cells not constitutively overexpressing PD-L1 due to oncogenic mutation, PD-L1 expression levels can act as a surrogate marker of the presence of activated T cells that recognize cognate tumour antigen expression on cancer cells and release interferon-γ [59]. Consistent with that, interferon-γ treatment of melanoma cells results in an increased level of PD-L1 in the melanoma cell EVs [31]. As interferon-γ results in PD-L1 expression by surrounding cells that express surface interferon receptors, PD-L1 expression is frequently detected on non-cancer cells such as lymphocytes and macrophages in the tumour microenvironment, and on T cells, with PD-L1 expression being equal or higher in the normal immune cells compared to the tumour cells [60,61,62]. In this study, PD-L1 was detected in all EV samples and, while there was no significant difference in PD-L1 levels between MM and healthy EVs, we did observe that PD-L1 was significantly higher in the EVs of MM patients responding to DARA treatment relative to patients progressing. This could suggest that patients responding to DARA treatment have high amount of infiltrating activated T cells that are upregulating PD-L1 in these patients. This hypothesis is supported by our mass spectrometry finding that the T cell marker CD8a was significantly higher in EVs from DARA responders relative to patients progressing (Figure 6).

Of the currently available CD38-antibodies, DARA is the most effective inducer of complement-dependent cytotoxicity (CDC) [63]. The complement inhibitory molecules CD55 and CD59 have been shown to be increased on BM cells collected at the time of progression from patients undergoing monotherapy with DARA [11]. Furthermore, neutralization of CD59 with the CD59 inhibitor, recombinant ILYd4, sensitized ARH-77 myeloma cells to the CDC effect mediated by rituximab (20 µg/mL) in a dose-dependent manner [64]. In this study, we reveal that CD55 and CD59 are present at significantly higher levels in MM PB EVs relative to healthy controls EVs (Figure 2). Indeed, CD59 levels were higher on both MM PB and BM EVs compared to healthy controls. We were not able to detect a difference in the CD55 and CD59 levels when comparing responders to non-responders (Figure 3). Nijhof et al. found that the expression of CD55 and CD59 on BM MM cells increases during treatment with DARA, although this increase is only significant when comparing pre-treatment BM samples with BM samples collected at progression [11]. This correlates with our results, where we found a significant difference when comparing NDMM to patients who have received DARA (Figure 4). Overall, the finding of high levels of CD59 on circulating plasma EV provides new evidence that MM plasma EVs reflect the BM cell and EV phenotype.

In addition to CD59, we also found that the EV marker CD147 was elevated in MM EVs relative to healthy controls EVs and further elevated in BM EVs relative to PB EVs. CD147 stimulates the production of matrix metalloproteinases and expression of VEGF, which contributes to angiogenesis [65,66]. The CD147 ligand CypA has been shown to promote signalling changes, migration, and proliferation of MM cells and the inhibition of CD147 with an anti-CD147 antibody supressed migration, tumour growth, and BM-colonization in a mouse xenograft model of MM [67]. Co-culture experiments revealed that tumour cell interactions with macrophages resulted in increased expression of CD147 and induction of MMP-9 and VEGF, and CD147 levels have a positive correlation with M2 Tumour-associated macrophage (TAM) infiltration and negative correlation with MM patient survival [66,68]. CD147 gene expression was shown to be significantly higher in MM compared with normal plasma cells, MGUS and SMM and higher CD147 gene expression is associated with poor prognosis in MM [69,70]. Thus, the MM EV CD147 profile outlined in this study is consistent with overexpression in MM cells and may represent a novel non-invasive biomarker that reflects the MM cellular phenotype.

The group of responders and non-responders is very heterogeneous. The prior treatment that they have received could have affected the results. Furthermore, the non-responders have received a median of 4 prior lines compared to the responders, who have received 2 prior lines. It would have been more accurate to include only patients who had received a specific number and type of prior treatment, although this would have decreased the number of participating patients significantly. However, our aim was to describe EVs from patients receiving DARA regardless of prior treatment. Although our group of NDMM contained only four patients, we found significant differences in the levels of complement inhibitory proteins (Figure 4). This could be a true finding that may become more significant if more patients were added to the NDMM group or, alternatively, these four NDMM patients by chance had lower levels of CD55 and CD59, and adding more patients to the group would eliminate the difference. A future study with a larger number of patients would clarify the issue. In addition, a limitation to the study is that we do not have BM samples from the responders due to lack of ethical approval for sample collection.

In conclusion, the exploitation of EVs as fluid-based cancer biomarkers has the potential to revolutionise cancer treatment through regular access to patient molecular information over the course of the disease and treatment. One example is the FDA approved non-invasive EV based urine test ExoDx Prostate IntelliScore (EPI Test, Bio-Techne). In the case of myeloma, molecular information is currently obtained through invasive bone marrow aspirate and/or biopsy for plasma cell molecular profiling. In this study, we reveal a similar pattern of CD38 on patient derived PB and BM EVs suggesting that PB EVs reflect MM BM plasma cells following DARA treatment. In addition, we reveal that MM PB and BM EV CD147, and complement inhibitory molecules CD55 and CD59 levels may have diagnostic and prognostic value, whereas EV PD-L1 levels may indicate response to DARA therapy. Further studies with longitudinal sampling are required to identify the most informative sampling time after initiation of therapy, to determine the maximum clinical benefit of the MM EV signature identified through this study.

## Figures and Tables

**Figure 1 cells-11-03365-f001:**
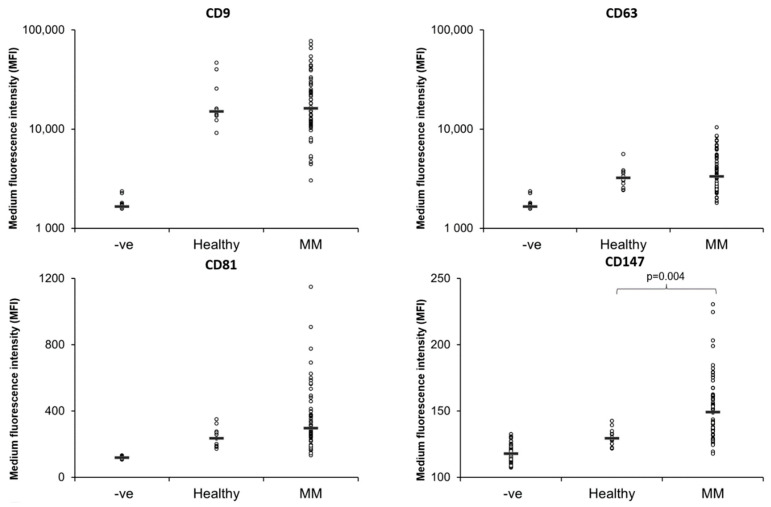
Healthy and Multiple Myeloma peripheral blood EVs are positive for EV markers; CD9, CD63, CD81 and CD147. 1.25 × 10^7^ EVs/test from 61 MM patients and 12 healthy controls were bound to the surface of 4 µm aldehyde/sulfate latex beads and stained with antibodies for the EV markers; CD9, CD63, CD81 and CD147 for 30 min on ice. The samples were then resuspended in 200 µL PBS and flow cytometry analysis was performed on a CytoFLEX LX Flow Cytometer. Gating of EV-decorated 4 μm in diameter beads was performed based on FCS/SSC parameters, so that unbound EVs or possible antibody aggregates are excluded from the analysis.

**Figure 2 cells-11-03365-f002:**
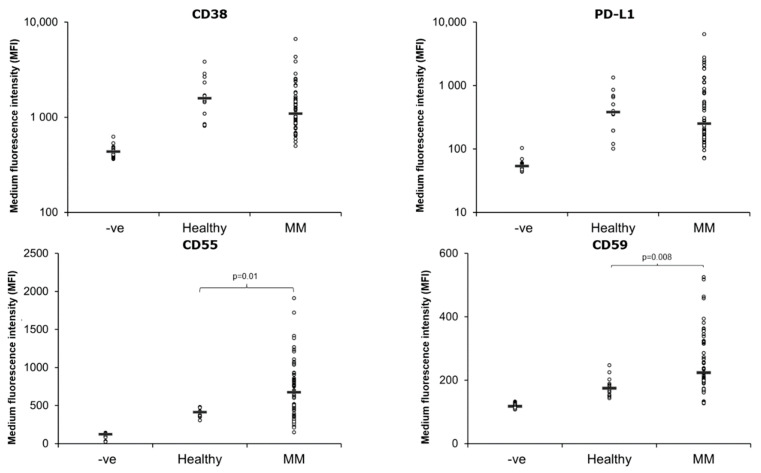
Complement inhibitory proteins CD55 and CD59 are higher on Multiple Myeloma peripheral blood EVs relative to healthy control PB EVs. 1.25 × 10^7^ EVs/test from 61 MM patients and 12 healthy controls were bound to the surface of 4 µm aldehyde/sulfate latex beads and stained with antibodies for CD38 and PD-L1 and the complement inhibitory proteins CD55 and CD59 for 30 min on ice. The samples were then resuspended in 200 µL PBS and flow cytometry analysis was performed on a CytoFLEX LX Flow Cytometer. Gating of EV-decorated 4 μm in diameter beads was performed based on FCS/SSC parameters, so that unbound EVs or possible antibody aggregates are excluded from the analysis.

**Figure 3 cells-11-03365-f003:**
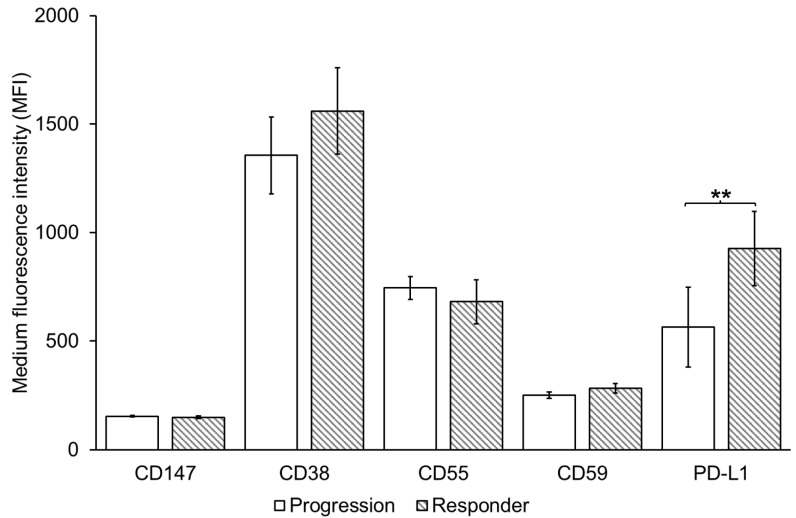
PD-L1 is higher on Multiple Myeloma peripheral blood EVs from patients responding to DARA relative to patients progressing on DARA. 1.25 × 10^7^ EVs/test from 19 responders and 38 patients progressing on DARA were bound to the surface of 4 µm aldehyde/sulfate latex beads and stained with antibodies for CD38 and PD-L1 and the complement inhibitory proteins CD55 and CD59 for 30 min on ice. The samples were then resuspended in 200 µL PBS and flow cytometry analysis was performed on a CytoFLEX LX Flow Cytometer. Gating of EV-decorated 4 μm in diameter beads was performed based on FCS/SSC parameters, so that unbound EVs or possible antibody aggregates are excluded from the analysis. Error bars refer to standard error of the mean (** *p* < 0.01 by two-tailed unpaired Student’s *t*-test).

**Figure 4 cells-11-03365-f004:**
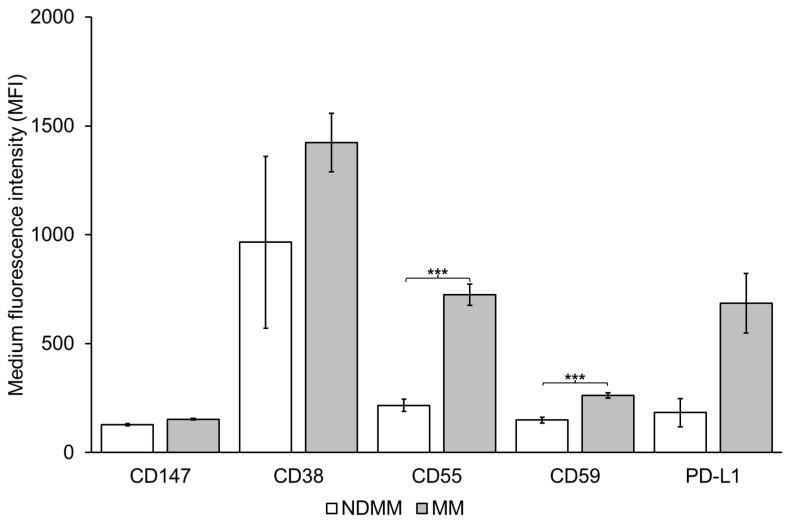
Complement inhibitory proteins CD55 and CD59 are higher on Multiple Myeloma (MM) peripheral blood EVs relative to newly diagnosed Multiple Myeloma (NDMM) PB EVs. 1.25 × 10^7^ EVs/test from 4 NDMM and 57 MM patients were bound to the surface of 4 µm aldehyde/sulfate latex beads and stained with antibodies for CD38 and PD-L1 and the complement inhibitory proteins CD55 and CD59 for 30 min on ice. The samples were then resuspended in 200 µL PBS and flow cytometry analysis was performed on a CytoFLEX LX Flow Cytometer. Gating of EV-decorated 4 μm in diameter beads was performed based on FCS/SSC parameters, so that unbound EVs or possible antibody aggregates are excluded from the analysis Error bars refer to standard error of the mean (*** *p* < 0.001 by two-tailed unpaired Student’s *t*-test).

**Figure 5 cells-11-03365-f005:**
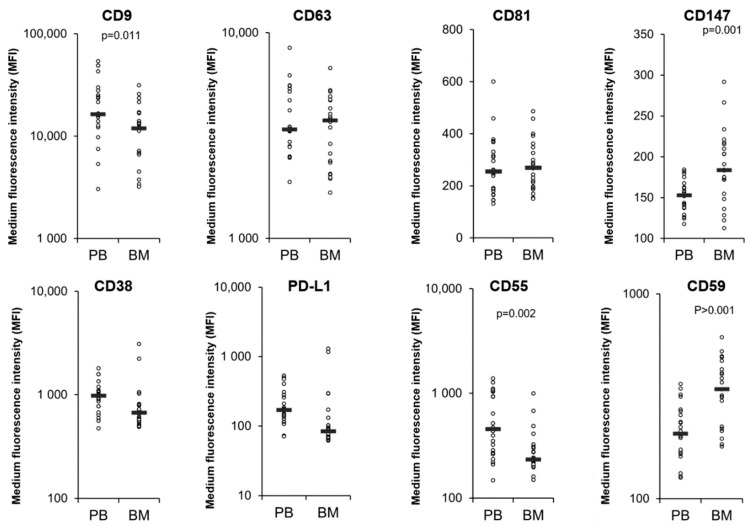
Multiple Myeloma bone marrow EVs have elevated CD59 and CD147 and decreased CD55 and CD9 levels compared to peripheral blood EV, whereas CD38 and PD-L1 are similar across all EV samples. 1.25 × 10^7^ EVs/test from 23 matched BM and PB EVs from patients progressing on DARA were bound to the surface of 4 µm aldehyde/sulfate latex beads and stained with antibodies for; CD9, CD63, CD81, CD147 CD38, PD-L1 and the complement inhibitory proteins CD55 and CD59 for 30 min on ice. The samples were then resuspended in 200 µL PBS and flow cytometry analysis was performed on a CytoFLEX LX Flow Cytometer. Gating of EV-decorated 4 μm in diameter beads was performed based on FCS/SSC parameters, so that unbound EVs or possible antibody aggregates are excluded from the analysis.

**Figure 6 cells-11-03365-f006:**
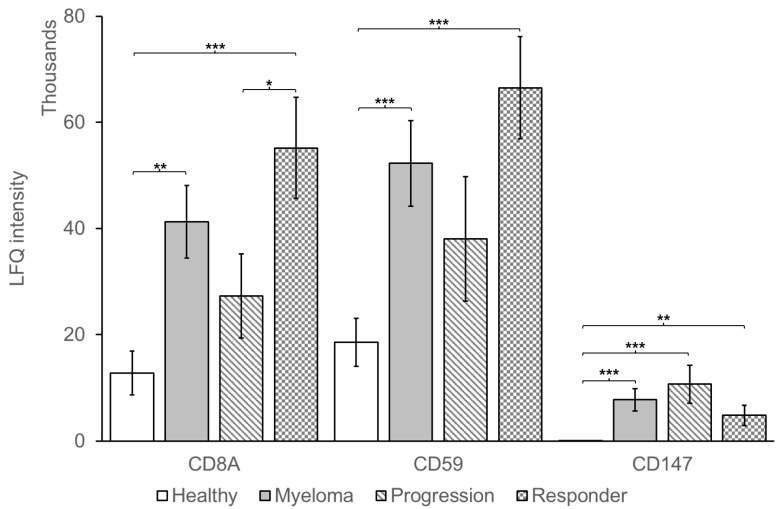
Differentially regulated proteins identified in multiple myeloma patient plasma EVs relative to healthy patient plasma EVs by LC-MS/MS. 2.5 × 10^8^ EV isolates from 10 MM patients (5 responders and 5 patients progressing on DARA) and 10 healthy controls were resuspended in 6 M urea, 50 mM Tris-HCl, reduced and alkylated using dithiothreitol and iodoacetamide. Following trypsin digestion each sample analyzed in duplicate on a Bruker timsTof Pro mass spectrometer connected to an Evosep One liquid chromatography system. Error bars refer to standard error of the mean (* *p* < 0.05, ** *p* < 0.01 and *** *p* < 0.001 by two-tailed unpaired Student’s *t*-test).

**Table 1 cells-11-03365-t001:** Patient characteristics.

Patient Characteristics	Responders n = 19	Non-Responders n = 38
**Age** *; years; median (range)	61 (49–83)	66 (47–84)
**Sex**; n (%)		
Female	7 (37)	20 (53)
Male	12 (63)	18 (47)
**Immunoglobulin subtype** *; n (%)		
IgG	12 (63)	25 (66)
IgA	1 (5)	3 (8)
Light-chain only	6 (32)	9 (23.5)
Non secretory	0 (0)	1 (2.5)
**ISS** *; n(%)		
I	5 (26)	13 (34)
II	8 (42)	8 (21)
III	4 (21)	5 (13)
Unknown	2 (11)	12 (32)
**ECOG performance status** *; n (%)		
0	9 (47)	20 (53)
1	7 (37)	6 (16)
2	0 (0)	0 (0)
3	0 (0)	2 (5)
Unknown	3 (16)	10 (26)
**Fluorescence in situ hybridization** ^§^; n (%)		
High-risk	1 (5)	9 (24)
Standard-risk	15 (79)	19 (50)
Unknown	3 (16)	10 (26)
**Number of prior lines of therapy**; median (range)	2 (1–5)	4 (0–17)

* = at the time of diagnosis of multiple myeloma. ^§^ = If assessed more than once, the most recent result prior to initiation of DARA treatment is shown. High-risk aberrations were defined by the presence of either t(4;14), t(14;16) or del17p, each detected with a cut-off of 10% according to national standards for cytogenetic evaluation [45].

**Table 2 cells-11-03365-t002:** Detection of DARA peptide sequences by mass spectrometry on MM PB EVs.

	DARA-Specific Sequences only Detected in DARA Treated MM Patient EVs									
MM Patient	1	2	3	4	5	6	7	8	9	10
**Light chain**		SNWPPTFGQGTKVEIKRTVAAPSVFIFPPSDEQLKSGTASVVCLLNNFYPR		SLEPEDFAVY	SNWPPTFGQGTKVEIKRTVAAPSVFIFPPSDEQLKSGTASVVCLLNNFYPR	SNWPPTFGQGTKVEIKRTVAAPSVFIFPPSDEQLKSGTASVVCLLNNFYPREAKVQWKVDNALQSGNSQESVTEQDSK				
**Heavy chain**	STSGGT		SGGTAA		GPSVFPLAPSSKSTSGGTAALGCLVK	GGTAAL	GPSVFPLAPSSKSTSGGTAALGCLVK	SCDKTHTCPPCPAPELLGGPSVFLFPPKPKDTLMISR		TKGPSVF
	KAKGQP				SGVHTF	ALTSGV				STKGPSVFPLA
					KPSNTK	TYICNVNHK				KGPSVF
					VEPKSCDK	SCDKTHTCPPCPAPELLGGPSVFLFPPKPKDTLMISR				HKPSNTKVD
						PSVFLFP				EEMTKNQVSLTCLVKGFYPSDIAVEWESNGQPENNYK

The DARA peptides that were detected in MM patient EVs, containing sequences not found in healthy control EVs, are listed for each patient with the unique amino acids underlined.

## Data Availability

The data presented in this study are available in “Extracellular Vesicles Isolated from Plasma of Multiple Myeloma Patients Treated with Daratumumab Express CD38, PD-L1, and the Complement Inhibitory Proteins CD55 and CD59” and the Appendix A.

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
