# Peer review of "Extracellular Vesicles Isolated from Plasma of Multiple Myeloma Patients Treated with Daratumumab Express CD38, PD-L1, and the Complement Inhibitory Proteins CD55 and CD59"

_cells, 2022, doi:10.3390/cells11213365_

Round 1
Reviewer 1 Report
The study by Brennan et al is a manuscript with high scientific standards. The study aimed to evaluate the role of CD38, CD55, CD59 and PD-L1 in MM patients on DARA treatment, The results and conclusions show acceptable data. However, there are a few suggestions for the authors:
1) The introduction section needs more literature regarding these markers in studies on responders vs, non-responders. The studies should ideally be related to DARA but if there is significance of these markers in other treatment types in MM , it would help support the case of the authors . It will also help the readers build a foundation on the reason for conducting this study.
2) The study design is good. However, can you clarify why only 4 patients with newly diagnosed MM were taken into the study? A larger sample size of NDMM would have given a powerful statistical impact . Also the role of markers in comparison of NDMM and responders vs. non-responders would have been quite interesting.
3) The BM and simultaneous PB samples were taken for non-responders and NDMM. I understand there may be an ethical issue associated with taking BM from responding patients, however, this sampling would have helped the authors objectives i.e. show the role of EV markers as a surrogate for bone marrow. Kindly clarify in the limitations about this
4) The prior treatment lines for responders and non-responders was 2 and > 4 respectively. Can you clarify in the discussion if the prior treatment lines may have had any impact on the up and down regulation of the markers ? Also clarify the time line sample collection ( meaning identify how much gap was given between the sample collection and prior treatment lines). Its possible that the impact of treatment lines may be showing up and down regulation of these markers and must be addressed for the reader. Also, prior treatment lines must be mentioned in the methods section
5) Please clarify why CD147 was significant upregulated in EVs of MM patients as compared to healthy controls? The mechanism of action of CD147 may be associated with this and should be addressed in the manuscript for better understanding of this EV marker.
6)The role of IFN gamma in PD-L1 expression is documented . Kindly clarify if IFN gamma was tested in the EVs ? This would have given a wider basis for PD-L1 expression. If this was tested , please put data ( even if non-significant). However, if this was not tested, can the authors test it and put the results.
7) In the Western blot gel, APOB is not clearly seen. Please clarify why this is observed.
Please add limitations of the study at the end of the manuscript
Overall the study is well written and the data presented is of high standard
Reviewer 2 Report
The manuscript by Brennan and Iversen et al., investigates the surface expression of extracellular vesicles (EV) Isolated from plasma of multiple myeloma patients treated with Daratumumab. The manuscript is well written and concerns an important concept of potentially using EVs as a predictive biomarker. However, the study design with relatively few patients and without longitudinal measurements prohibits many potential insights and makes it very hard to draw any clear conclusions.
Major revision:
1. The hypothesis of the study and the reason of the somewhat strange study design should be clarified. As the authors mention in the discussion longitudinal measurements would very much improve the study. The reason for the difference of subjects in the study groups are also unclear. Also, the inclusion of only 4 newly diagnosed patients makes the purpose of including this group very unclear.
2. The inclusion of BM EVs here is very unclear, especially since there is no comparison to healthy subjects. I would consider removing this from the story or clarify its relevance.
3. Data presentation. Much of the important data are shown in tables instead of figures (eg table 2 and 4), which would make it much easier for the reader to interpret the data. Also, the significance of the data in table 4 is unclear.
4. Unclear experimental designs. The authors discuss and seem interested in the hypothesis of CD38+ EVs acting as decoy as part of DARA resistance. However, there was no difference in the levels of CD38 between healthy subjects and MM patients (rather an unexpected tendency for higher levels in healthy) or between responders vs non-responders. The authors should consider improving the experimental set up to investigate this in a better way so this could be shown or disputed. Why didn’t they for instance compare the difference between responders and non-responders with LC-MS/MS (instead of only taking 5/19 responders and 5/38 non-responders and putting these together vs healthy subjects)?
5. Unclear conclusions. Regarding PDL1, the authors suggest (in the discussion) “patients responding to DARA treatment have high amount of infiltrating activated T cells that are upregulating PD-L1 in these patients. This hypothesis is supported by our mass spectrometry finding that the T cell marker CD8a was significantly higher in EVs from DARA responders (table 4).” Are the findings in table 4 significant? How does this hypothesis go together with the findings that there is no difference in PD-L1 EV expression compared to healthy subjects (Fig 2 )?
6. The authors defined the expression of the EV markers using only Mean Fluorescent Intensity (MFI). I would suggest that also the percentage expression of the markers is shown. There are no figures showing gates for EVs and beads comparing with or without antibodies. In supplementary figure 2A, the authors showed a plot but do not show the gating strategy of which population was actual used. In figure no. 3 the MFI of CD81 was very low compared to CD9 and CD63, can the authors show the percentage of these three EVS markers in both PB and BM?
Minor revision:
1. The section title 3.3 is used twice (p. 9 and p. 11), also on page 9 the section title includes BM EVs, however this section only discusses PB EVs.
2. The authors used 2.5X108 EVs for Mass spectrometry, can the authors state what was the concentration of protein in 2.5X108 EVs?
Reviewer 3 Report
The paper by Brennan et al reports on observations coming from a previous report (13) focused on the ability of the therapeutic Daratumumab antibody to modify the myeloma membrane to produce extra cellular vesicles (ev). The paper reports on the results obtained by testing the expression of selected markers by ev, potentially involved in the immune regulation and escape of the tumor (antibody resistance). I am convinced that this observation may be of interest for a large audience of people working in the field of myeloma therapy and resistance to antibody treatment. I have only a comment on reference 10, which states that Daratumumab acts on the enzymatic function of the target molecule. The proper reference is: Therapeutic Opportunities with Pharmacological Inhibition of CD38 with Isatuximab. Martin TG, Corzo K, Chiron M, Velde HV, Abbadessa G, Campana F, Solanki M, Meng R, Lee H, Wiederschain D, Zhu C, Rak A, Anderson KC. Cells. 2019 Nov 26;8(12):1522. doi: 10.3390/cells8121522. PMID: 31779273
Author Response
The paper by Brennan et al reports on observations coming from a previous report (13) focused on the ability of the therapeutic Daratumumab antibody to modify the myeloma membrane to produce extra cellular vesicles (ev). The paper reports on the results obtained by testing the expression of selected markers by ev, potentially involved in the immune regulation and escape of the tumor (antibody resistance). I am convinced that this observation may be of interest for a large audience of people working in the field of myeloma therapy and resistance to antibody treatment. I have only a comment on reference 10, which states that Daratumumab acts on the enzymatic function of the target molecule. The proper reference is: Therapeutic Opportunities with Pharmacological Inhibition of CD38 with Isatuximab. Martin TG, Corzo K, Chiron M, Velde HV, Abbadessa G, Campana F, Solanki M, Meng R, Lee H, Wiederschain D, Zhu C, Rak A, Anderson KC. Cells. 2019 Nov 26;8(12):1522. doi: 10.3390/cells8121522. PMID: 31779273
Response: Thank you for the positive response. The reference has been changed.
Round 2
Reviewer 2 Report
I'm happy with the updated version and the respons from the authors.